# Skew Class-Balanced Re-Weighting for Unbiased Scene Graph Generation

**Haeyong Kang and Chang D. Yoo ***

School of Electrical Engineering, Korea Advanced Institute of Science and Technology (KAIST),
Daejeon 34141, Republic of Korea
* Correspondence: cd_yoo@kaist.ac.kr

**Abstract:** An unbiased scene graph generation (SGG) algorithm referred to as Skew Class-Balanced Re-Weighting (SCR) is proposed for considering the unbiased predicate prediction caused by the long-tailed distribution. The prior works focus mainly on alleviating the deteriorating performances of the minority predicate predictions, showing drastic dropping recall scores, i.e., losing the majority predicate performances. It has not yet correctly analyzed the trade-off between majority and minority predicate performances in the limited SGG datasets. In this paper, to alleviate the issue, the *Skew Class-Balanced Re-Weighting* (SCR) loss function is considered for the unbiased SGG models. Leveraged by the skewness of biased predicate predictions, the SCR estimates the target predicate weight coefficient and then re-weights more to the biased predicates for better trading-off between the majority predicates and the minority ones. Extensive experiments conducted on the standard Visual Genome dataset and Open Image V4 and V6 show the performances and generality of the SCR with the traditional SGG models.

**Keywords:** scene graph generation (SGG); skew class-balanced re-weighting (SCR); predicate sample estimates; skew class-balanced effective number





## 1. Introduction

Currently, computer vision [1–6] has been prevalent in various fields: material [7], chemical [8,9], and medical science [10]. Of great interest are tasks related to classifying and detecting objects by classical neural networks [11–15]. With the development of computer vision, graph neural networks [16–22] and multimodal text and image models [23–28] have also been intensely investigated. However, there is less information on Scene Graph Generation (SGG) models in the literature. For this reason, SGG has recently been receiving increased attention for improving image comprehension, divided into two approaches: one-stage [29,30] and two-stage [31–45]. The core building blocks of SGG are the objects in the image. There can be diverse relationships among the objects [46,47]. All relationships can be represented as triplets *subject, relation, object*, which can be used for generating the scene graph. The precise interpretation of an image depends on the core relations between the *subject–object* pairs. For example, given an image of *dog, woman, kite* objects, the image can be interpreted in multiple ways such as ⟨*dog, has, leg*⟩, ⟨*dog, near, woman*⟩, and ⟨*woman, holding, kite*⟩.

In general, however, the real-world large data sets often have shown long-tailed predicate distributions as depicted in Figure 1a—the predicate proportion of the Visual Genome [31], containing 51 predicates. One cause of long-tail distribution is the background predicates. The background predicates account for more than 80% in total predicates since it is not trivial to annotate all possible relationships, leading to possibly far more *subject-object* pairs without annotating ground truth than ones with ground truth labels. Another reason is the visual event frequency, as shown in Figure 1b. The majority predicates more frequently occur than the minority ones given a visual *subject–object* pair in the real-world scene. For example, given a *man-board* pair, the possible predicates—*backgrounds, on, holding,*

*riding, etc.* represents a long-tail distribution. The predicate *on* is more frequently observed than *holding* and *riding*.

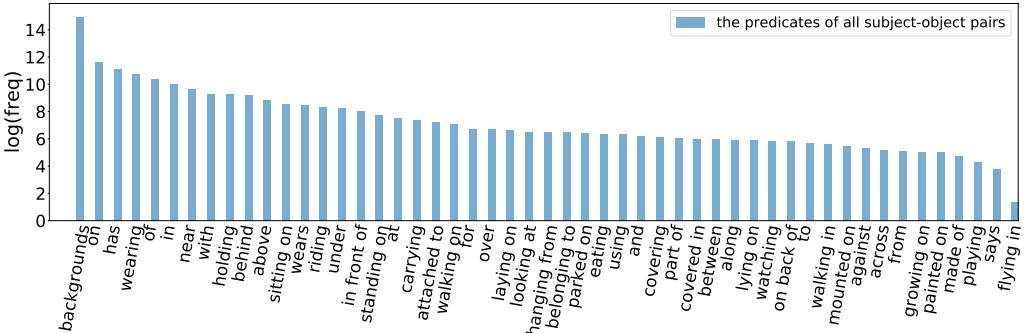

(**a**) The long-tailed predicate distribution.

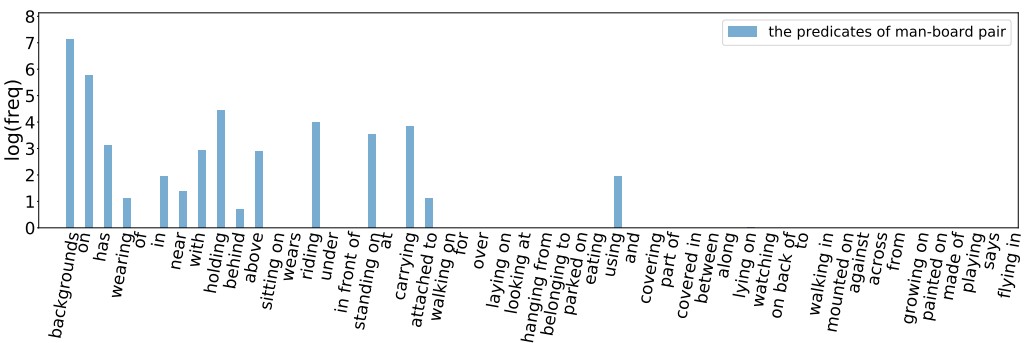

(**b**) The predicate distribution of the *subject–object* pair.

**Figure 1.** The long-tailed predicate distribution: (**a**) the predicate proportion of the visual genome [31], where backgrounds are possibly far more than others and (**b**) the possible predicates of a *man-board* pair.

This long-tailed label distribution causes the trained model to tend to be biased toward the majority predicates. The SGG models [43,48–52] focus on increasing the minority predicate performances for the unbiased scene graph generation, i.e., increasing mean recall scores. However, these methods lead to deteriorating performances of the majority predicates, drastically dropping recall scores, i.e., losing the majority predicate performances. Recently, the issue has been alleviated through resampling images and object instances [49]. However, this method needs more computing power and time to train the model on the resampled samples. The current state-of-the-art models focus on re-balancing biased loss [53] or correcting noisy labels [54] to acquire an unbiased SGG model. Nevertheless, the prior works have not yet adequately described and analyzed the trade-off performances between majority and minority predicates for learning SGG models based on the given imbalanced datasets.

In this paper, the *Skew Class-Balanced Re-Weighting* (SCR) loss function is considered for alleviating the issues and acquiring the best trade-off performances in the multiple SGG tasks. Leveraged by the skewness of predicate predictions, the SCR estimates its weight coefficients and then reweights more to biased predicate samples to adaptively be unbiased SGG models. The extensive experimental results show that the SCR loss function gives more generalized performances than priors in the multiple SGG tasks on the Visual Genome dataset [55] and the Open Images datasets [56].

**Contributions of our SCR** learning scheme to unbiased SGG models:

- Leveraged by the skewness of biased predicate predictions, the *Skew Class-Balanced Re-Weighting* (SCR) loss function is firstly proposed for the unbiased scene graph generation (SGG) models.
- The SCR is applied to the current state-of-the-art SGG models to show its effectiveness, leading to more generalized performances: the SCR outperforms the prior reweighted methods on both mean recall and recall measurements in the multiple SGG tasks.

This paper is organized as follows. The Related Work section provides discussions on unbiased scene graph generation. The unbiased SGG section presents scene graph generation. In the Skew Class-Balanced Re-Weighting (SCR) section, the SCR loss function is depicted in detail. In the experimental section, the results of scene graph generation on the Visual Genome and Open Image V4 and V6 dataset are examined, along with an analysis and ablation study on the SCR with the current state-of-the-art SGG models. Finally, we conclude.

## 2. Related Works

**Unbiased Scene Graph Generation (SGG)**. Predicate distribution is much more long-tailed than object distribution. For $N$ objects and $R$ predicates, the model has to address the fundamental challenge of learning $O(N^2R)$ relations with few [57,58]. To overcome the limited training dataset, the linguistic external knowledge [38,57,59] was used by Yu et al. [60], regularizing the deep neural network; using linguistic knowledge, the probabilistic model has also alleviated the semantic ambiguity of visual relationships [61]. Furthermore, to alleviate the imbalanced relationship distribution, Yin et al. [38] reformulated the conventional one-hot classification as a *n*-hot multiclass hierarchical recognition via novel Intra-Hierarchical trees (IH-trees) for each label set in the triplet $\langle subject, predicate, object \rangle$. Recently, unbiased SGG [43,47–54,62–71] has drawn unprecedented interest for more generalized SGG models. Occurrence-based Node Priority Sensitive (NPS)-loss [47] was used for balancing predictions; the Total Direct Effect (TDE) method has proposed firstly for unbiased learning by Tang et al. [43], which directly separates the bias from biased predictions through the counterfactual methodologies on causal graphs; CogTree [48] addressed the debiasing issue based on the coarse-to-fine structure of the relationships from the cognition viewpoint; Li et al. [49] improved the context modeling for tail categories by designing the bipartite graph network and message propagation on resampled objects and images. Lastly, the Predicate Probability Distribution based Loss (PPDL) [53] has proposed to train the biased SGG models, which measure the semantic predicate representation to re-balance the biased training loss. In this work, the *Skew Class-Balanced Re-Weighting* (SCR) loss function is proposed for alleviating biased predicate predictions, leading to the most generalized SGG models through the novel adaptive re-weighting learning scheme.

**Re-Weighting Based Unbiased SGG.** Overall unbiased SGG models can be categorized into the re-balancing strategy of re-weighting [47,48,50,53] and re-sampling [49] and biased model-based strategy [43,51,52]. For unbiased SGG modes, Tang et al. [43] first investigated the re-weighting learning algorithm. However, they observed that the performances of the majority predicates were drastically dropped, resulting in low recall scores while with high mean recall scores. This shows the general tendency that there is a trade-off performance between majority predicates and minority ones. To alleviate the trade-off issue, Yu et al. [48] proposed CogTree based on the coarse-to-fine structure of the relationships from the cognition viewpoint. Recently, the Predicate Probability Distribution (PPD) [53] re-balances the biased training loss according to the similarity between the predicted probability distribution and the estimated one. However, it has not yet correctly analyzed the trade-off performances between the majority and minority classes in various SGG tasks. In this paper, we measure the sample skew score based on the sample estimates for bias toward the majority classes to assign the sample weight correctly. The sample skewness is computed as the Fisher–Pearson coefficient of skewness on its sample mean value [72]. However, since the mean value tends to be biased toward the majority predi-

cates, we measure the sample skew score fairly on its target logit instead of its mean value. Based on the sample skew score, the SCR assigns the sample weights adaptively—if there is no bias, we assign fewer weights to the sample (majority). If it is biased to one side, we assign larger weights to the samples (minority). Such that the SGG models with SCR show superior performances and generality on the multiple SGG tasks.

### 3. Unbiased Scene Graph Generation

In this section, we discuss the general scene graph generation model of the object and predicate predictions and depict the predicate sample estimates for measuring its skewness of biased predictions.

*3.1. Scene Graph Generations*

Given an image $I$, a scene graph model generates a graph $\mathcal{G} = (\mathcal{V}, \mathcal{E})$, where $\mathcal{V}$ and $\mathcal{E}$ are the sets of nodes and edges, respectively. Each node $o_i \in \mathcal{V}$ is represented by a bounding box $v^{bbox} \in \mathbb{R}^4$ and a corresponding class label $o_y \in \mathcal{C}_{obj}$. Each edge $r_{i,j} \in \mathcal{E}$ represents the predicate between the subject node $v_i$ and the object node $v_j$. The corresponding predicate label is $r_y \in \mathcal{C}_{rel}$.

We depict the proposed unbiased SGG framework as shown in Figure 2. Object detector outputs its predictions (a): *man* and *hourse*. Given *man-horse* label prediction pair, the traditional SGG model predicts (d) predicate predictions ($\hat{R}$ see Equation (3)) through a fusion of (b) non-visual FREQ prior predicates and (c) visual predicate predictions ($\hat{R}_{vis}$). For unbiased SGG model training, our SCR estimates (e) a sample size of the possible predicate candidates through FREQ ($R_{freq}$ or $R_{emb}$) and measures (f) the target label skew score and then calculates (g) the training target sample weight for the adaptive re-weighted loss. In (f) and (g), red lines around the target label indicate predicate skew ($S^i_{skew}$ see Equation (9)) and re-weight scores ($W^i$ see Equation (7)), respectively, where they have an approximately inverse relationship.

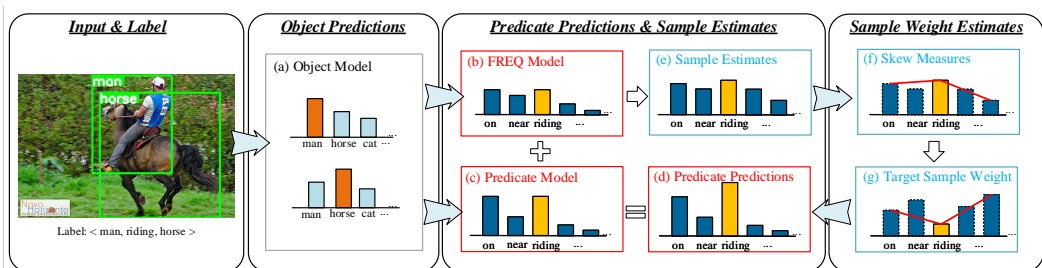

**Figure 2.** Skew class-balanced re-weighting (SCR) for unbiased SGG models. Based on (**a**) object detection, the SGG model predicts (**d**) predicates of (**b**) FREQ and (**c**) visual predictions; unbiased SGG is trained by (**g**) target sample weight, which is estimated by (**f**) skew measures of (**e**) possible sample estimates.

3.1.1. Object Predictions

Following [43], the node features are derived from the object detector. In particular, for each bounding box $v^{bbox}_i$, the detector returns a RoI-Align feature $x^{RoI}_i$ and an object label embedding $l_i$. In general SGG model, the $\mathcal{N}$ number of node features are constructed by vector concatenation $X = \{[x^{RoI}_i; l_i; b_i]\}^{\mathcal{N}}_{i=1}$, where $b_i$ is the embedded box feature from the box coordinate $v^{bbox}_i$.

$$\tilde{X} = fc_{obj}(X) \in \mathbb{R}^{\mathcal{N} \times \mathcal{D}_{obj}}, \qquad (1)$$

where all $fc_*$ denote a fully connected layer for linear transformations or logits, and the object feature dimension $\mathcal{D}_{obj}$ depends on the SGG model. As depicted in Figure 2a, the predicted object label of $\hat{O} \in \mathbb{R}^{\mathcal{N} \times |\mathcal{C}_{obj}|}$ is estimated by object logits $fc(\tilde{X})$ as follows:

$$\hat{O} = fc(\tilde{X}). \qquad (2)$$

### 3.1.2. Predicate Predictions

Inspired by Tang et al. [43], Zhu et al. [73], predicate predictions can be made by employing multiple logits from visual and non-visual features. We follow the sum over all outputs to generate the final predicate prediction. As illustrated in Figure 2d, the combined predicate logit $\hat{R}$ is estimated based on the summation of the visual logits and the non-visual logits as follows:

$$\hat{R} = \hat{R}_{vis} \oplus \hat{R}_{freq} \oplus \hat{R}_{emb}, \tag{3}$$

where $\hat{R} \in \mathbb{R}^{\mathcal{N}(\mathcal{N}-1) \times |\mathcal{C}_{rel}|}$; $\oplus$ is an element-wise sum; as shown in Figure 2c, $\hat{R}_{vis}$ is the predicate logits from visual feature $F_{vis}$ such as a $\mathcal{D}_{vis}$ dimensional union feature and a *subject–object* pair feature, which also depends on the SGG model,

$$\begin{aligned} \hat{R}_{vis} &= fc_{vis}(F_{vis}), \quad F_{vis} \in \mathbb{R}^{\mathcal{N}(\mathcal{N}-1) \times \mathcal{D}_{vis}}, \\ \hat{R}_{freq} &= \text{Sigmoid}(R_{freq}) \in \mathbb{R}^{\mathcal{N}(\mathcal{N}-1) \times |\mathcal{C}_{rel}|}, \\ \hat{R}_{emb} &= fc_{emb}(L_{emb}), \quad L_{emb} \in \mathbb{R}^{\mathcal{N}(\mathcal{N}-1) \times 400}. \end{aligned} \tag{4}$$

The FREQ [74] as a non-visual feature, $R_{freq}$ (Figure 2b) looks up the empirical distribution over relationships between subject $\hat{o}_i$ and object $\hat{o}_j$ as computed in the training set where $\hat{o}_i, \hat{o}_j \in \hat{O}$. However, since FREQ does not consider any image representations when predicting predicates, it tends to lead to biased predicate predictions due to its imbalanced predicate distribution. To minimize the biased effects, we use the Sigmoid-activated FREQ predicate logits. In addition, for acquiring the more smoothness of the empirical distribution, the concatenated *subject–object* embedding $\hat{R}_{emb}$ is added to the predicate predictions where $L_{emb} = \{[l_i; l_j]\}_{i,j}^{\mathcal{N}}$.

### 3.2. Sample Estimates

Non-visual predicate features tend to be more biased than visual features due to an imbalanced training set. If the degree of biased predictions can be measured, we can leverage its value to learn the SGG models without bias. According to Brown [72], the degree of bias prediction can generally be measured in a skew score. In general, if there is no bias, the skew score is close to 0, and if it is biased to one side, the skew score is over either $-1$ or $+1$.

In SCR, we need to estimate how many predicate samples are biased to measure the predicate skew scores. To approximate the biased sample numbers, we use two non-visual prior predicate distributions of FREQ, $\hat{R}_{freq}$ and *subject–object* label embedding $\hat{R}_{emb}$ as described in Figure 2e. Based on the two predicate distributions, at first, we define the skew logit $\hat{R}_{skew}$ with the Sigmoid activation function as follows:

$$\hat{R}_{skew} = \text{Sigmoid}(R_{freq} \oplus \hat{R}_{emb}). \tag{5}$$

To estimate the predicate sample weights properly, the SCR approximates skewness through the predicate sample estimates $\hat{R}_{skew}$ in Equation (5). We investigate the best predicate sample estimates through several experiments with a combination of non-visual predictions as follows:

- SCR of EMB: $\hat{R}_{skew} = \text{Sigmoid}(\hat{R}_{emb})$;
- SCR of FREQ: $\hat{R}_{skew} = \text{Sigmoid}(R_{freq})$;
- SCR of FREQ+EMB: $\hat{R}_{skew} = \text{Sigmoid}(R_{freq} \oplus \hat{R}_{emb})$.

Then, the predicate sample estimates are acquired as follows:

$$\mathcal{M} = \sum_{i}^{\mathcal{N}(\mathcal{N}-1)} \hat{r}_i, \tag{6}$$

where $\hat{r}_i \in \hat{R}_{skew}$; the estimated $m_y \in \mathcal{M}$ is the number of $y$th predicate sample size and $m_y \in [0, \mathcal{N}(\mathcal{N}-1))$.

Moreover, the skew predicate $\hat{R}_{skew}$ serves to estimate the predicate candidates that a *subject–object* pair can have fairly since the Sigmoid activation function suppresses the more extensive biased predictions. For example, a *Man-Horse* pair may have predicates such as *riding, on, with,* etc. The Sigmoid activation function amplifies the frequency of the minority predicates while squeezing that of the majority ones, as shown in Figure 2b,e, which is used to calculate Skew Class-Balanced Effective Number that we depict in the following section.

Note in Equation (5), the Sigmoid activation function was applied to estimate the predicate sample sizes (Equation (6)) and measure skewness (Equation (9)), given a subject–object pair. The main reason was that it provides smoothly activated outputs from 0 to 1. We assumed that an output value of 1 is regarded as one predicate.

## 4. Skew Class-Balanced Re-Weighting (SCR)

In this section, leveraged by the biased predictions deduced by the predicate sample estimates, the *Skew Class-Balanced Re-Weighting* (SCR) performs sample weight estimates for learning unbiased SGG models.

### 4.1. Skew Class-Balanced Effective Number

The *Skew Class-Balanced Effective Number* approximates the mini-batch class-balanced re-weighting coefficients $E_{m_y}$ based on the predicate skew logit $\hat{R}_{skew}$ in Equation (5). The $i$th predicate sample $E_{m_y}^i$ is defined as follows as [75,76]:

$$E_{m_y}^i = \frac{1 - \beta_i^{m_y}}{(1 - \beta_i)}, \tag{7}$$

where $\beta_i = (m_y - 1)/m_y$; the effective number satisfies the following properties that $E_{m_y}^i = 1$ if $\beta_i = 0$ $(m_y = 1)$; $E_{m_y}^i \to m_y$ as $\beta_i \to 1$ $(m_y \to \infty)$ such that $\beta_i$ controls how fast $E_{m_y}^i$ grows as the target predicate sample size $m_y$ increases.

To estimate the $i$th predicate effective number $E_{m_y}^i$, we adaptively estimate the $\beta_i \in [0, 1)$ by using the entropy $H_{skew}^i$ and skew score function $S_{skew}^i$ of the $\hat{R}_{skew}$ as shown in Algorithm 1: if $S_{skew}^i > S_{th}$ then the $\beta_i \in [0, 1)$ that assigns more weights to the minority predicate samples than the majority ones; otherwise, $S_{skew}^i \leq S_{th}$ then $\beta_i = 0$ that re-weights uniformly over the entire class sample loss, i.e., conventional cross-entropy loss. The threshold $S_{th}$ is determined as follows:

$$S_{th} = \bar{S}_{skew} - \delta, \tag{8}$$

where $\bar{S}_{skew}$ and $\delta = 0.7$ are the mean value of $S_{skew}$ and the hyper-parameter used, respectively, in all experiments.

### 4.2. Skew Measures

In training, the predicate sample skewness depends on the predicate label. In other words, according to Brown [72], some majority predicate skew values tend to be greater than zero while others tend to be zero or negative. This is an unfair skew measure, leading the SCR Algorithm 1 to more weights either in the majority predicates or in the minority ones. In this paper, as portrayed in Figure 2f, to measure the skew value of all target labels fairly, we use the target skew logit instead of the mean value. Therefore, the $i$th predicate sample skew $S_{skew}^i$ is firstly measured by the following equation given the target label index $y$ as follows:

$$S_{skew}^i = \frac{\frac{1}{|\mathcal{C}_{rel}|} \sum_{\hat{r}_{i,j} \in \hat{R}_{skew}^i} (\hat{r}_{i,j} - \hat{r}_{i,y})^3}{\left( \frac{1}{|\mathcal{C}_{rel}|} \sum_{\hat{r}_{i,j} \in \hat{R}_{skew}^i} (\hat{r}_{i,j} - \hat{r}_{i,y})^2 \right)^{3/2}}. \tag{9}$$

Then, we use the uniformness to determine the $\beta$. The uniformness provides confidence in the predicate sample estimates. To calculate the uniformness of the $i$th predicate sample, we estimate the entropy $H^i_{skew}$ as follows:

$$H^i_{skew} = -\lambda_{skew} \sum_{\hat{r}_{i,j} \in \hat{\mathbf{R}}^i_{skew}} p(\hat{r}_{i,j}) \log_{|\mathcal{C}_{rel}|} p(\hat{r}_{i,j}), \qquad (10)$$

where the number of predicates $|\mathcal{C}_{rel}|$ is used as the base of the logarithm, which $H^i_{skew} \in [0,1]$; the $\hat{r}_i \in \hat{\mathbf{R}}_{skew}$ have uniform distributions when $H^i_{skew}$ is close to 1; otherwise, the predicate sample may have either skew distributions or correct distributions and the $\lambda_{skew} = 0.06$ is the coefficient of $S_{skew}$. The following section depicts the relationship between skew and entropy in detail.

---

**Algorithm 1** Skew Class-Balanced Effective Number.

---

**Require:** Dataset $\mathcal{D}$, SGG Model $f_\theta$

1: **for** $t = 0, 1, 2, \ldots, T$ **do**
2:      $\mathcal{B} \leftarrow \text{Minibatch}(\mathcal{D})$
3:      $\hat{\mathbf{R}}_{skew} \leftarrow Skew\_Logits(\hat{\mathbf{R}}_{freq}, \hat{\mathbf{R}}_{emb}; \mathcal{B}, f_\theta)$          (Equation (5))
4:      $\mathcal{M} \leftarrow Sample\_Estimates(\hat{\mathbf{R}}_{skew})$          (Equation (6))
5:      $S_{skew} \leftarrow Skew(\hat{\mathbf{R}}_{skew}, \mathbf{R})$          (Equation (9))
6:      $H_{skew} \leftarrow Entropy(\hat{\mathbf{R}}_{skew})$          (Equation (10))
7:      **if** $S_{skew} > S_{th}$ **then**
8:          $\beta = \mathbf{1.0} - H_{skew}$
9:      **else**
10:          $\beta = \mathbf{0.0}$
11:      **end if**
12:      $E^i_{m_y} = \frac{1 - \beta_i^{m_y}}{1 - \beta_i}; m_y \in \mathcal{M}, i \in [0, \mathcal{N}(\mathcal{N} - 1))$          (Equation (7))
13: **end for**

---

### 4.3. Target Sample Weights

The interpretation of the relationship between skew and weight is depicted based on biased predicate prediction as shown in Figure 3. In Figure 3a, to understand the sample weight estimate from the predicate sample estimates, we assume that the predicate biased predictions $\hat{\mathbf{R}}_{skew}$ are given by the predicate sample frequency $\mathcal{M}^{1/5}$ (see Figure 1a) and the predicate entropy $H_{skew}$, i.e., the more frequent sample is the more biased prediction is; the more biased the prediction is the less entropy is. The skew $S_{skew}$ measures not only the degree of true/false prediction but also biased predictions as shown in Figure 3c. The following simple equation summarizes the degree of true/false predictions:

$$\begin{cases} S^i_{skew} > 0 & \text{if } \arg\max_j \hat{r}_{i,j} \neq r_y \\ S^i_{skew} < 0 & \text{otherwise.} \end{cases} \qquad (11)$$

where $S^i_{skew} \approx -1$ for the true prediction; $S^i_{skew} = 0$ for the uniform prediction and $S^i_{skew} \gg 0$ for the false and biased prediction. Moreover, the predicate target label-wise skew has more discrete at high entropy. The final $i$th sample weight $w^i_y = 1/E^i_{m_y}$ (Figure 2g) is acquired by the criteria of skew and entropy as shown in Figure 3b. The minority predicates have larger weights than the majority when $S^i_{skew} \gg 0$.

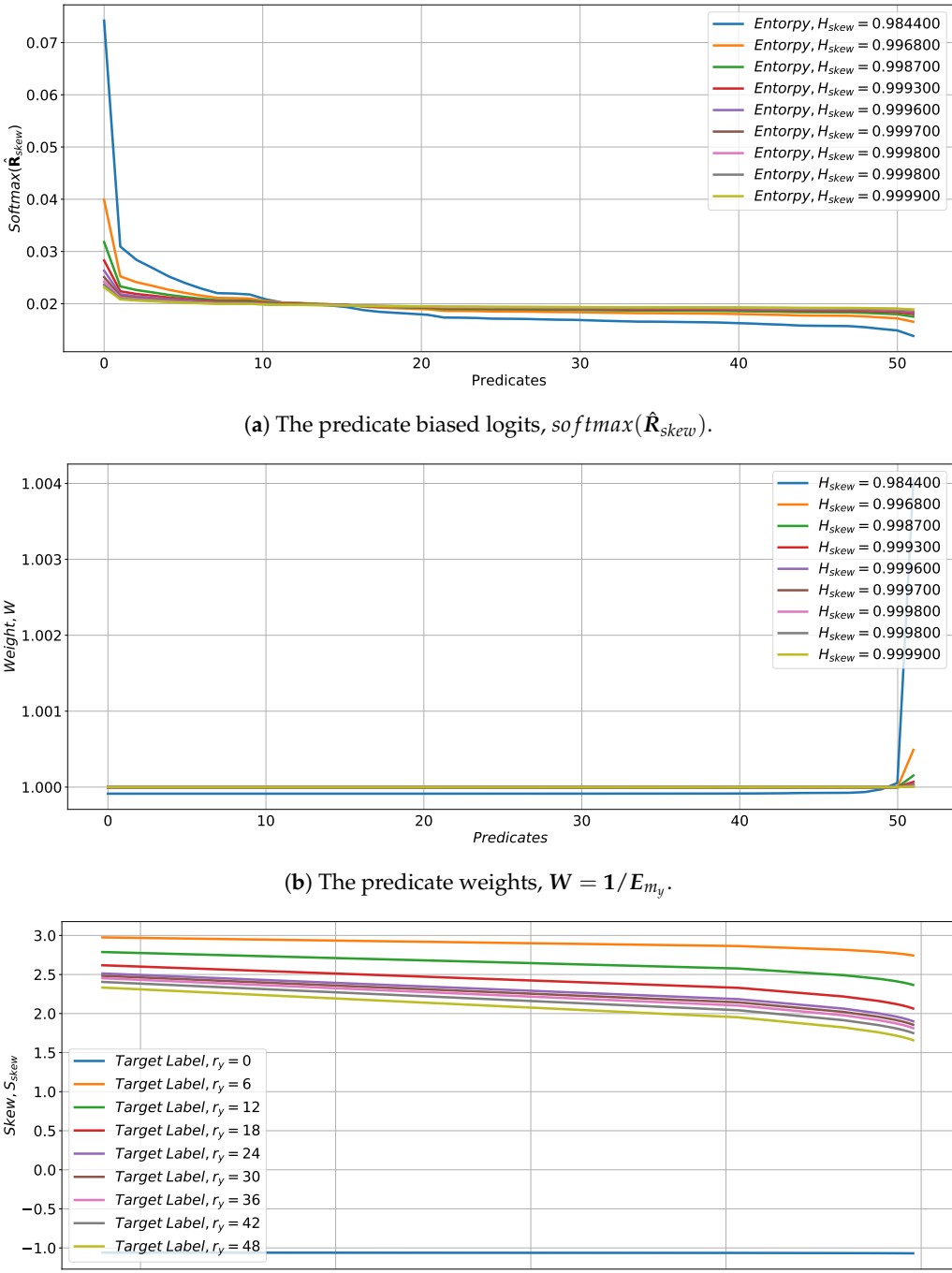

(**a**) The predicate biased logits, $softmax(\hat{R}_{skew})$.

(**b**) The predicate weights, $W = 1/E_{m_y}$.

(**c**) The predicate target label-wise skew, $S_{skew}$

**Figure 3.** The biased predicate weights: (**a**) the predicate biased logits $\hat{R}_{skew}$ of label frequency $\mathcal{M}^{1/5}$, depending on entropy $H_{skew}$, (**b**) the sensitivity of sample weights $W$, according to entropy $H_{skew}$, and (**c**) the skew measures $S_{skew}$ of a target label $r_y$.

*4.4. Learning with SCR*

Except for object loss, all traditional SGG models are learned by the skew class-balanced re-weighting cross-entropy loss functions. The conventional cross-entropy loss for the objects is computed, given object predictions $\hat{O}$ and the object ground-truth $o_y^i \in O$ as follows:

$$\mathcal{L}_{obj}(\hat{\boldsymbol{O}}, \boldsymbol{O}) = \sum_{i}^{\mathcal{N}} -\gamma_{obj} \cdot w_y^i \log\left(\frac{\exp(\hat{o}_y)}{\sum_{\hat{o}_j \in \hat{\boldsymbol{O}}_i} \exp(\hat{o}_j)}\right), \tag{12}$$

where, given $\mathcal{N}$ of objects, $i$th object sample weight is $w_y^i = 1$ s.t. $|\mathcal{C}_{obj}| = \sum_j^{|\mathcal{C}_{obj}|} w_j^i$; here, the sample weight denominator $\gamma_{obj} = 1/\sum_i^{\mathcal{N}} w_y^i$ is used for the mean object cross-entropy loss. The skew class-balanced cross-entropy loss computes the skew predicate-balanced cross-entropy, based on the predicate predictions $\hat{\boldsymbol{R}}$ and the predicate ground-truth $r_y^i \in \boldsymbol{R}$:

$$\mathcal{L}_{rel}(\hat{\boldsymbol{R}}, \boldsymbol{R}) = \sum_{i}^{\mathcal{N}(\mathcal{N}-1)} -\gamma_{rel} \cdot w_y^i \log\left(\frac{\exp(\hat{r}_y)}{\sum_{\hat{r}_j \in \hat{\boldsymbol{R}}_i} \exp(\hat{r}_j)}\right), \tag{13}$$

where, given $\mathcal{N}(\mathcal{N}-1)$ of object-subject pairs, the $i$th predicate sample weight is $w_y^i = \frac{1}{E_{m_y}^i} = \frac{1-\beta_i}{1-\beta_i^{m_y}}$ s.t. $|\mathcal{C}_{rel}| = \sum_j^{|\mathcal{C}_{rel}|} w_j^i$; here, we set the predicate sample normalizer as $\gamma_{rel} = 1/\sum_i^{\mathcal{N}(\mathcal{N}-1)} w_y^i$ for mean predicate loss. In summary, the total objective loss function $\mathcal{L}_{total}$ for unbiased SGG learning can be formulated as follows:

$$\mathcal{L}_{total} = \mathcal{L}_{obj} + \mathcal{L}_{rel}. \tag{14}$$

## 5. Experiments

The proposed SCR is evaluated with the traditional SGG models on the Visual Genome benchmark datasets [55], and the performances of the SCR are compared with others in the multiple SGG tasks. The source code is available at https://github.com/ihaeyong/Unbiased-SGG (accessed on 26 January 2023).

### 5.1. Visual Genome

We used Visual Genome (VG) [55] dataset to train and evaluate our models, which is composed of 108k images across 75k object categories and 37k predicate categories. We followed the widely adopted VG split [31,42,74] containing the most frequent 150 object categories and 50 predicate categories. The original split only has a training set (70%) and a test set (30%). We followed Zellers et al. [74] to sample a 5k validation set from the training set for parameter tuning.

### 5.2. Open Images

The Open Images dataset [56] is a large-scale dataset proposed by Google recently. Compared with the Visual Genome dataset, it has a superior annotation quality for the scene graph generation. In this work, we conduct experiments on Open Images *V4&V6*, following similar data processing and evaluation protocols in [47,56,77]. The Open Images V4 is introduced as a benchmark for scene graph generation by Zhang et al. [77] and Lin et al. [47], which has 53,953 and 3234 images for the train and validation sets, 57 objects categories, and 9 predicate categories in total. The Open Images V6 has 126,368 images used for training, 1813, and 5322 images for validation and testing, respectively, with 301 object categories and 31 predicate categories. This dataset has a comparable amount of semantics categories with the VG.

### 5.3. Experiments Configurations

**State-of-the-Art Comparisons.** For fair comparisons, all the compared SGG models should use the FREQ [74], which looks up the empirical distribution over relationships between subject prediction $\hat{o}_i$ and object ones $\hat{o}_j$. To evaluate the effectiveness of the SCR learning algorithm, we follow the same experimental settings as the CogTree [48]. We also set the current state-of-the-art SGG models as the baseline: MOTIFS [74], VCTree [41] and SG-Transformer [48] which contains 3 *O2O* blocks and 2 *R2O* blocks with 12 attention

heads, and Bipartite-Graph [49] without resampling layers and compare the performance with the state-of-the-art debiasing approach TDE [43], CogTree [48], PCPL [50], DLFE [51], BPL-SA [52], PPDL [53], and NICE [54].

**Implementation.** Following the previous works [43,48,49], the object detector is the pre-trained Faster R-CNN [78] with ResNeXt-101-FPN [79]. In bi-level resampling [49], we also set the repeat factor $t = 0.07$, instances drop rate $\gamma_d = 0.7$, and weight of fusion the entities features $\rho = -5$. The $\alpha, \beta$ are initialized as 2.2 and 0.025, respectively.

### 5.4. Evaluations

Our SCR has the following two evaluations:

**Relationship Retrieval (RR)** contains three sub-task: (1) Predicate Classification (PredCls): taking ground truth bounding boxes and labels as inputs, (2) Scene Graph Classification (SGCls): using ground truth bounding boxes without labels, (3) Scene Graph Detection (SGDet): detecting SGs from scratch. The conventional metric of RR is **Recall@K (R@K)**, included in this paper even though the biased prediction is reported by Misra et al. [80] for the performance of the SCR. Moreover, to evaluate the general performances, we adopted **mean Recall@ K (mR@K)** that retrieves each predicate separately and then averages R@K for all predicates.

**Zero-Shot Relationship Retrieval (ZSRR)**. The **Zero-Shot Recall@K** was firstly evaluated on the VG dataset in [43], which reports the R@K of those subject-predicate-object triplets that have never been observed in the training set. ZSRR also has three sub-tasks as RR.

### 5.5. Quantitative Results

**Visual Genome.** The SCR is compared with others on the two evaluation tasks: RR and ZSRR, which are the same as shown in Tables 1 and 2. The SCR achieves the best and second best performances over the previous methods: TDE, PCPL, Cogtree, DLFE, BPL-SA PPDL, and NICE, demonstrating its generality and effectiveness on the two measures of RR task. Moreover, the SCR shows the best trade-off performances on the ZSRR task as shown in Figure 2.

**The Best Trade-off performances.** To analyze the trade-off between majority and minority predicate performances, firstly, we need to understand the two measurements, mRR and RR. The mRR was introduced to calculate the recall on each predicate category independently and then to average the results by Tang et al. [41]. This is because of RR's bias, which was reported by Misra et al. [80]. The higher the recall scores (RR) are, the more biased the majority categories are. Therefore, a higher mRR is a more unbiased SGG model; a higher RR is a more biased SGG model. Traditional SGG models show the tendency that if mRR is higher than RR, RR performances are low, i.e., MOTIFS, VCTREE, and SG-Transformer show the best Recall but low mRR@100 in Table 1. However, our SCR shows the best trade-off in terms of mRR and RR measurements; MOTIFS with SCR shows the best performances in mRR and the second-best performances in RR except for MOTIFS. So do the other models' results. Figure 4 supports our analysis of the best trade-off since overall predicate performances were better than the traditional SGG model (SG-Transformer).

**Open Image V4 and V6.** To show the effectiveness of SCR, we set BGNN as the baseline, as shown in Table 3. On Open Images Dataset V4, SCR outperformed BGNN except for $score_{wtd}$ measurement. Mainly, SCR shows outstanding performance in terms of phrase evaluation. Moreover, on Open Images Dataset V6, SCR outperformed all baselines such as BGNN, GPS-Net, Etc., showing a good trade-off between mean recall@50 and recall@50. These results proved that we could have a good trade-off performance in long-tailed predicated distributions if properly assigning weights in training.

**Table 1.** The SGG performances of relationship retrieval on mean Recall@K and Recall@K. SCR$^\dagger$ denotes SCR of FREQ+EMB. Note the best and second best methods under each setting are marked according to format.

| | PredCls | | SGCls | | SGDet | |
|---|---|---|---|---|---|---|
| Model | mR@20/50/100 | R@20/50/100 | mR@20/50/100 | R@20/50/100 | mR@20/50/100 | R@20/50/100 |
| IMP+ [31] | -.-/ 9.8/10.5 | 52.7/59.3/61.3 | -.-/ 5.8/ 6.0 | 31.7/34.6/35.4 | -.-/ 3.8/ 4.8 | 14.6/20.7/24.5 |
| FREQ [74] | 8.3/13.0/16.0 | 53.6/60.6/62.2 | 5.1/ 7.2/ 8.5 | 29.3/32.3/32.9 | 4.5/ 6.1/ 7.1 | 20.1/26.2/30.1 |
| KERN [46] | -.-/17.7/19.2 | -.-/65.8/67.6 | -.-/ 9.4/10.0 | -.-/36.7/37.4 | -.-/ 6.4/ 7.3 | -.-/27.1/29.8 |
| MOTIFS [74] | 10.8/14.0/15.3 | 58.5/65.2/67.1 | 6.3/ 7.7/ 8.2 | 32.9/35.8/36.5 | 4.2/ 5.7/ 6.6 | 21.4/27.2/30.3 |
| VCTree [41] | 14.0/17.9/19.4 | 60.1/66.4/68.1 | 8.2/10.1/10.8 | 35.2/38.1/38.8 | 5.2/ 6.9/ 8.0 | 22.0/27.9/31.3 |
| MSDN [33] | -.-/15.9/17.5 | -.-/64.6/66.6 | -.-/ 9.3/ 9.7 | -.-/38.4/39.8 | -.-/ 6.1/ 7.2 | -.-/31.9/36.6 |
| G-RCNN [37] | -.-/16.4/17.2 | -.-/64.8/66.7 | -.-/ 9.0/ 9.5 | -.-/38.5/37.0 | -.-/ 5.8/ 6.6 | -.-/29.7/32.8 |
| BGNN [49] | -.-/30.4/32.9 | -.-/59.2/61.3 | -.-/14.3/16.5 | -.-/37.4/38.5 | -.-/10.7/12.6 | -.-/31.0/35.8 |
| DT2-ACBS [44] | -.-/35.9/39.7 | -.-/23.3/25.6 | -.-/24.8/27.5 | -.-/16.2/17.6 | -.-/22.0/24.4 | -.-/15.0/16.3 |
| MOTIFS [74] | 11.5/14.6/15.8 | 59.5/66.0/67.9 | 6.5/ 8.0/ 8.5 | 35.8/39.1/39.9 | 4.1/ 5.5/ 6.8 | 25.1/32.1/36.9 |
| + TDE [43] | 18.5/25.5/29.1 | 33.6/46.2/51.4 | 9.8/13.1/14.9 | 21.7/27.7/29.9 | 5.8/ 8.2/ 9.8 | 12.4/16.9/20.3 |
| + PCPL [50] | -.-/24.3/26.1 | -.-/54.7/56.5 | -.-/12.0/12.7 | -.-/35.3/36.1 | -.-/10.7/12.6 | -.-/27.8/31.7 |
| + CogTree [48] | 20.9/26.4/29.0 | 31.1/35.6/36.8 | 12.1/14.9/16.1 | 19.4/21.6/22.2 | 7.9/10.4/11.8 | 15.7/20.0/22.1 |
| + DLFE [51] | 22.1/26.9/28.8 | -.-/52.5/54.2 | 12.8/15.2/15.9 | -.-/32.3/33.1 | 8.6/11.7/13.8 | -.-/25.4/29.4 |
| + BPL-SA [52] | 24.8/29.7/31.7 | -.-/50.7/52.5 | 14.0/16.5/17.5 | -.-/30.1/31.0 | 10.7/13.5/15.6 | -.-/23.0/26.9 |
| + PPDL [53] | -.-/32.2/33.3 | -.-/47.2/47.6 | -.-/17.5/18.2 | -.-/28.4/29.3 | -.-/11.4/13.5 | -.-/21.2/23.9 |
| + NICE [54] | -.-/29.9/32.3 | -.-/55.1/57.2 | -.-/16.6/17.9 | -.-/33.1/34.0 | -.-/12.2/14.4 | -.-/27.8/31.8 |
| **+ SCR$^\dagger$ (ours)** | 25.9/31.5/33.6 | 51.0/57.9/60.1 | 14.2/17.1/18.2 | 27.1/31.0/32.3 | 9.6/13.5/15.9 | 18.1/25.1/29.5 |
| VCTree [41] | 11.7/14.9/16.1 | 59.8/66.2/68.1 | 6.2/ 7.5/ 7.9 | 37.0/40.5/41.4 | 4.2/ 5.7/ 6.9 | 24.7/31.5/36.2 |
| + TDE [43] | 18.4/25.4/28.7 | 36.2/47.2/51.6 | 8.9/12.2/14.0 | 19.9/25.4/27.9 | 6.9/ 9.3/11.1 | 14.0/19.4/23.2 |
| + PCPL [50] | -.-/22.8/24.5 | -.-/56.9/58.7 | -.-/15.2/16.1 | -.-/40.6/41.7 | -.-/10.8/12.6 | -.-/26.6/30.3 |
| + CogTree [48] | 22.0/27.6/29.7 | 39.0/44.0/45.4 | 15.4/18.8/19.9 | 27.8/30.9/31.7 | 7.8/10.4/12.1 | 14.0/18.2/20.4 |
| + DLFE [51] | 20.8/25.3/27.1 | -.-/51.8/53.5 | 15.8/18.9/20.0 | -.-/33.5/34.6 | 8.6/11.8/13.8 | -.-/22.7/26.3 |
| + BPL-SA [52] | 26.2/30.6/32.6 | -.-/50.0/51.8 | 17.2/20.1/21.2 | -.-/34.0/35.0 | 10.6/13.5/15.7 | -.-/21.7/25.5 |
| + PPDL [53] | -.-/33.3/33.8 | -.-/47.6/48.0 | -.-/14.3/15.7 | -.-/32.1/33.0 | -.-/11.3/13.3 | -.-/20.1/22.9 |
| + NICE [54] | -.-/30.7/33.0 | -.-/55.0/56.9 | -.-/19.9/21.3 | -.-/37.8/39.0 | -.-/11.9/14.1 | -.-/27.0/30.8 |
| **+ SCR$^\dagger$ (ours)** | 27.7/33.5/35.5 | 49.7/56.4/58.3 | 15.4/18.9/20.1 | 26.7/30.6/31.9 | 10.3/13.8/16.3 | 18.1/25.0/29.4 |
| SG-Transformer [48] | 14.8/19.2/20.5 | 58.5/65.0/66.7 | 8.9/11.6/12.6 | 35.6/38.9/39.8 | 5.6/ 7.7/ 9.0 | 24.0/30.3/33.3 |
| + CogTree[48] | 22.9/28.4/31.0 | 34.1/38.4/39.7 | 13.0/15.7/16.7 | 20.8/22.9/23.4 | 7.9/11.1/12.7 | 15.1/19.5/21.7 |
| **+ SCR$^\dagger$ (ours)** | 27.0/32.2/34.5 | 45.3/52.7/55.0 | 14.9/17.7/18.7 | 25.1/28.9/30.2 | 10.4 /13.4/15.0 | 17.7/23.2/26.2 |

**Table 2.** The SGG Performances of zero-shot relationship retrieval on Recall@K. SCR$^\dagger$ denotes SCR of FREQ+EMB. The SGG models re-implemented under our codebase are denoted by the superscript *.

| Zero-Shot Relationship Retrieval | | PredCls | | SGCls | | SGDet | |
|---|---|---|---|---|---|---|---|
| Model | Method | R@50 | R@100 | R@50 | R@100 | R@50 | R@100 |
| MOTIFS [43] | baseline [43] | 10.9 | 14.5 | 2.2 | 3.0 | 0.1 | 0.2 |
| | Reweight [43] | 0.7 | 0.9 | 0.1 | 0.1 | 0.0 | 0.0 |
| | TDE [43] | 14.4 | 18.2 | 3.4 | 4.5 | 2.3 | 2.9 |
| | CogTree [48] * | 2.4 | 4.0 | 0.9 | 1.5 | 0.3 | 0.6 |
| | **SCR$^\dagger$ (ours)** | 18.0 | 21.1 | 5.1 | 5.9 | 2.4 | 3.8 |
| VCTree [43] | Baseline [43] | 10.8 | 14.3 | 1.9 | 2.6 | 0.2 | 0.7 |
| | TDE [43] | 14.3 | 17.6 | 3.2 | 4.0 | 2.6 | 3.2 |
| | CogTree [48] * | 3.3 | 5.0 | 2.1 | 2.6 | 0.4 | 0.6 |
| | **SCR$^\dagger$ (ours)** | 17.6 | 20.4 | 4.5 | 5.2 | 2.5 | 3.5 |
| SG-Transformer [48] * | Baseline * | 4.1 | 6.3 | 1.6 | 2.3 | 0.2 | 0.5 |
| | CogTree [48] * | 5.2 | 7.3 | 2.3 | 3.0 | 0.3 | 0.5 |
| | **SCR$^\dagger$ (ours)** | 16.4 | 19.6 | 4.6 | 5.3 | 2.0 | 3.2 |
| BGNN [49] | BGNN * | 15.0 | 18.0 | 4.5 | 5.4 | 4.5 | 5.3 |
| | CogTree [48] * | 13.4 | 16.1 | 5.0 | 5.7 | 0.5 | 0.8 |
| | **SCR$^\dagger$ (ours)** | 16.3 | 19.5 | 4.9 | 5.9 | 1.9 | 3.0 |

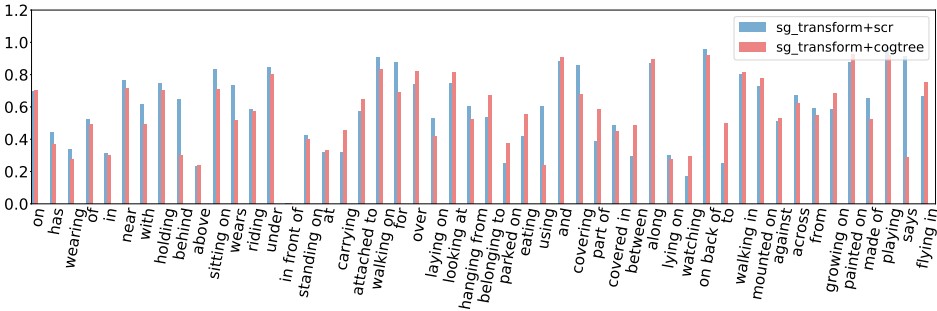

**Figure 4.** The Recall@100 on PredCls: we compare the SCR of FREQ+EMB with CogTree using the SG-Transformer [48].

**Table 3.** The performances of open images dataset. ∗ denotes results reproduced by Li et al. [49]. SCR† denotes SCR of FREQ+EMB.

| Dataset | Models | mR@50 | R@50 | wmAP | | score_wtd |
| | | | | rel | phr | |
|---------|--------|-------|------|-----|-----|-----------|
| V4 | RelDN [77] * | 70.40 | 75.66 | 36.13 | 39.91 | 45.21 |
| | GPS-Net [47] * | 69.50 | 74.65 | 35.02 | 39.40 | 44.70 |
| | BGNN [49] | 72.11 | 75.46 | 37.76 | 41.70 | **46.87** |
| | **BGNN+SCR† (ours)** | **72.20** | **75.48** | **38.64** | **45.01** | 45.01 |
| V6 | RelDN [77] * | 33.98 | 73.08 | 32.16 | 33.39 | 40.84 |
| | VCTree [41] * | 33.91 | 74.08 | 34.16 | 33.11 | 40.21 |
| | MOTIFS [74] * | 32.68 | 71.63 | 29.91 | 31.59 | 38.93 |
| | TDE [43] * | 35.47 | 69.30 | 30.74 | 32.80 | 39.27 |
| | GPS-Net [47] * | 35.26 | 74.81 | 32.85 | 33.98 | 41.69 |
| | BGNN [49] | 40.45 | 74.98 | 33.51 | 34.15 | 42.06 |
| | **BGNN+SCR† (ours)** | **42.43** | **75.21** | **33.98** | **35.13** | **42.66** |

## 5.6. Ablation Study

We investigate the predicate-biased prediction and the best hyper-parameter settings of the SCR loss function for the better generalized SGG models.

**Predicate Bias.** To estimate the predicate bias and assign the proper sample weights, we define the predicate sample estimates $\hat{R}_{skew}$ of $\hat{R}_{freq}$ and $\hat{R}_{emb}$ in Equation (5). To investigate the proper predicate sample estimates, we examined the effectiveness of the predicate sample estimates-SCR of EMB, SCR of FREQ, and SCR of FREQ+EMB with the fixed predicate predictions (Equation (3)) as shown in Table 4. In the experiments, the SCR of FREQ+EMB leads to more generalized performances over others. In summary, the previous Re-Weight methods worsen the recall performances, while the SCR does not.

**Table 4.** The ablation study for the predicate sample estimates on Recall@100. The underbar represents the predicate sample estimates for better generality.

| Relationship Retrieval | | PredCls | | | SGCls | | | SGDet | | |
| Model | Method | mRR | ZSRR | RR | mRR | ZSRR | RR | mRR | ZSRR | RR |
|------------------------|--------|---------|------|-----|-------|------|-----|-------|------|-----|
| VCTree | SCR of FREQ | 31.6 | 22.2 | 60.0 | 16.9 | 6.2 | 35.4 | 13.3 | 3.5 | 31.6 |
| | SCR of EMB | 33.3 | 21.1 | 60.1 | 17.9 | 5.9 | 35.2 | 15.4 | 3.5 | 30.0 |
| | **SCR of FREQ+EMB** | 33.7 | 20.1 | 60.0 | 18.0 | 6.1 | 33.3 | 14.4 | 3.5 | 31.9 |

**Hyper-parameters.** The hyper-parameters of SCR control the weight of the SGG loss. We investigate the best hyper-parameter settings as shown in Table 5. The best hyper-parameter settings $\delta = 0.7$ and $\lambda_{skew} = 0.06$ show the generalized performances of the SGG tasks. The smaller $\lambda_{skew}$ is, the higher mean Recall scores are, while the higher $\lambda_{skew}$ is, the higher Recall scores are, i.e., the $\lambda_{skew}$ controls the trade-off between the majority

predicates and the minority ones since the smaller $H_{skew}$ tend to assign more weights to the minority predicates in the SCR. The $\delta = 0.7$ shows the best proportion of Re-weighting the predicate samples.

**Table 5.** The ablation study for the scr hyper-parameters on Recall@100. The underbar represents the best trade-off performances between mRecall and Recall. SCR$^+$ denotes SCR of FREQ+EMB.

| Relationship Retrieval | | | PredCls | | SGCls | | SGDet | |
|---|---|---|---|---|---|---|---|---|
| Model | $\lambda_{skew}$ | $\delta$ | mRR | RR | mRR | RR | mRR | RR |
| VCTree +**SCR**$^+$ | 0.03 | 0.7 | **35.5** | 58.3 | **20.1** | 31.9 | **16.3** | 29.4 |
| | <u>0.06</u> | 0.7 | <u>33.7</u> | <u>60.0</u> | <u>18.0</u> | <u>33.3</u> | <u>14.4</u> | <u>31.9</u> |
| | 0.08 | 0.7 | 32.8 | 60.4 | 17.2 | 34.2 | 13.5 | 32.6 |
| | 0.06 | 0.6 | 33.5 | 60.6 | 17.8 | 34.3 | 14.2 | 32.0 |
| | 0.06 | <u>0.7</u> | <u>33.7</u> | <u>60.0</u> | <u>18.0</u> | <u>33.3</u> | <u>14.4</u> | <u>31.9</u> |
| | 0.06 | 0.8 | 33.3 | 59.3 | 17.6 | 33.0 | 14.0 | 30.6 |

### 5.7. Qualitative Examples

To demonstrate the effectiveness of the sample-wise SCR Re-weighting, we show the comparison of Recall @100 on PredCls of all predicates based on the SG-Transformer [48] as shown in Figure 4. The SCR of FREQ+EMB achieves a significant performance gain on the overall predicate categories. Moreover, the skew predicate $\hat{R}_{skew}$ serves to estimate not only the predicate candidates that a *subject–object* pair can have but also the target skew $S_{skew}$ which determines the sample weight. Figure 5 includes a *subject–object* pair which have possible predicates, its predicate prediction $\hat{R}_i$, and the target skew $S^i_{skew}$ deduced by the predicate sample estimate $\hat{R}^i_{skew}$. The higher the sample estimate is, the lower the target skew is. To be specific, when one *woman*- $S^{\text{wearing}}_{skew}$ -coat pair have possible predicates such as *backgrounds, has, wearing, in*, etc., we show predicate predictions $\hat{R}_i$ with its target skew $S^i_{skew}$ deduced by its sample estimates $\hat{R}^i_{skew}$. The larger $\hat{R}^i_{skew} \in \hat{R}_{skew}$ tends to be a smaller $S^i_{skew}$, when compared to another *hill*- $S^{\text{covered in}}_{skew}$ -snow pair in a SCR-trained SGG model.

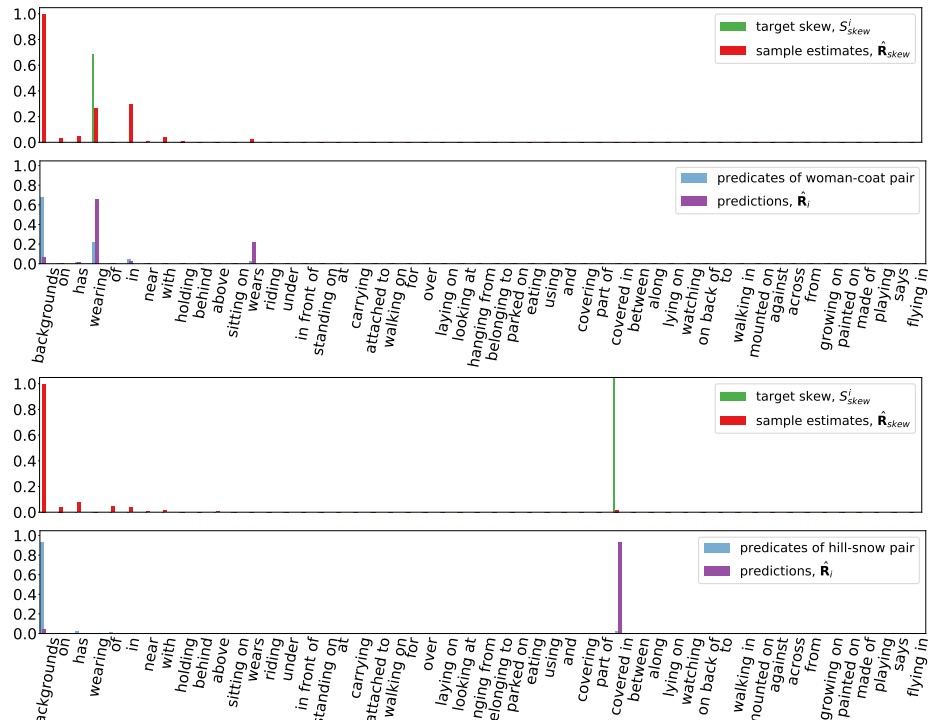

**Figure 5.** The predicate predictions and sample estimates: in a SCR-trained SGG model, the larger $\hat{R}^i_{skew}$ tends to be a smaller $S^i_{skew}$, i.e., *woman*- $S^{\text{wearing}}_{skew}$ -coat (**upper**) < *hill*- $S^{\text{covered in}}_{skew}$ -snow (**lower**).

## 6. Conclusions

In this paper, the unbiased Scene Graph Generation (SGG) algorithm, referred to as Skew Class-Balanced Re-Weighting (SCR), was proposed for considering the unbiased predicate prediction caused by the long-tailed distribution. The prior works focus mainly on alleviating the deteriorating performances of the minority predicate predictions, showing drastic dropping recall scores, i.e., forgetting the majority predicate class. It has not yet properly analyzed the trade-off performances between majority and minority predicates in the given SGG datasets. In this paper, to address the issues leveraged by the skewness of biased predicate predictions, firstly, the SCR estimated the predicate re-weighting coefficient and then re-weighted more to the biased predicates for the better trading-off performances between the majority and the minority predicates. Extensive experiments conducted on the standard Visual Genome dataset and Open Image V4 and V6 showed the SCR's effectiveness and generality with the traditional SGG models.

**Author Contributions:** Writing—original draft, H.K.; Writing—review & editing, C.D.Y. All authors have read and agreed to the published version of the manuscript.

**Funding:** This work was supported by the Institute of Information & communications Technology Planning & Evaluation (IITP) grant funded by the Korean government (MSIT) (No. 2022-0-00951, Development of Uncertainty-Aware Agents Learning by Asking Questions), and partly supported by the National Research Foundation of Korea (NRF) grant funded by the Korea government (MSIT) (No. 2022R1A2C2012706).

**Data Availability Statement:** Not applicable.

**Conflicts of Interest:** The authors declare no conflict of interest.

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
