# Peer review of "Skew Class-Balanced Re-Weighting for Unbiased Scene Graph Generation"

_make, doi:10.3390/make5010018_

Round 1

Reviewer 1 Report

In this paper, an unbiased scene graph generation (SGG) algorithm, called skew class balanced Reweighting (SCR), is proposed to take into account unbiased predicate prediction caused by long-tail distributions. In this paper, we consider the skew class balance reweighting (SCR) loss function of unbiased SGG models. Using the skewness of biased predicates predictions, the SCR estimates the weight coefficients of the target predicates and then reweights the biased predicates for a better trade-off between majority and minority predicates.

1) To what extent can the deviation be reduced with Sigmoid-activated FREQ predicate logic?

2) How to get a smoother distribution of experience by embedding the negative in the subject object and adding it to the predicate prediction.

3) Why can Sigmoid activation function inhibit a wider range of biased predictions?

4) Please elaborate on predicate weight estimation.

5)The explanation of equation 12 is not enough, What the symbol on the right side of the equation represents puzzles the readers.

6) Please elaborate on how the SCR loss function using the unbiased SGG model correctly analyzes the tradeoff between majority and minority predicate performance.

7) The authors ignore some relevant papers. For example, fusion methods that have been published in 2022 please see ("Brain tumor segmentation based on the fusion of deep   semantics and edge information in multimodal MRI" , Information Fusion ,Volume 91,  Pages 376-387.)  The authors should compare their method with it carefully.

Author Response

Thank you for your constructive feedback.

We prepared our responses to the reviewer’s questions. 

Please read our attached cover letter word file.

Reviewer 2 Report

The authors propose an algorithm for constructing unbiased forecasts. Experiments are carried out with data to generate graph scenes. The article may be useful to a certain circle of readers, however, for its publication, it is necessary to correct a number of comments:

1) Line 15: Section numbering should start from "1", not from "0".

2) Line 17: You need to detail the citation. Describe in more detail the works [1-15] by groups.

3) Authors can start the introductory part like this: “Currently, computer vision problems are very popular (https://doi.org/10.3390/s23031713, arXiv:1804.03928). Of great interest are tasks related to the recognition and detection of objects by classical neural networks (10.18287/2412-6179-CO-922, arXiv:1902.05888, doi.org/10.1016/j.jpdc.2016.09.001). But graph models (arXiv:2207.04396, 10.3390/math10214021, 0.1016/j.aiopen.2021.01.001) and multimodal text and image models (arXiv:2301.13823, 10.3390/s21041270) are also of great interest. However, there is less information on SGG models in the literature.

4) Line 70: Double quoting [28, 28, 29].

5) Figures 1-5: Figure titles are too long. It is better to make explanations in the text of the article.

6) The inscriptions in Figure 3 are hard to read. Try to enlarge them.

7) Check the coloring in table 1. For example, MOTIFS+PPDL in the second value in the first column is better than the proposed model, and is highlighted in blue.

8) It is necessary to describe in more detail table 1 and the results obtained.

Author Response

(The authors gave the same response as above.)

Round 2

Reviewer 2 Report

The authors eliminated all the comments, raising the level of the article to a sufficient one.